# Arterial Thrombosis in Patients with Acute Myeloid Leukemia: Incidence and Risk Factors

**DOI:** 10.3390/cancers15113060

**Published:** 2023-06-05

**Authors:** Mirjana Mitrovic, Nikola Pantic, Nikica Sabljic, Zoran Bukumiric, Marijana Virijevic, Zlatko Pravdic, Mirjana Cvetkovic, Jovan Rajic, Jelena Bodrozic, Violeta Milosevic, Milena Todorovic-Balint, Ana Vidovic, Nada Suvajdzic-Vukovic, Darko Antic

**Affiliations:** 1Clinic of Hematology, Unviersity Clinical Center of Serbia, 2 Koste Todorovica St., 11000 Belgrade, Serbia; drnikolapantic@gmail.com (N.P.); nsabljic19@gmail.com (N.S.); marijana.virijevic@yahoo.com (M.V.); zlatko.pravdic@gmail.com (Z.P.); mimamima.cvetkovic@gmail.com (M.C.); jovadin89@gmail.com (J.R.); bodrozic.jelena@gmail.com (J.B.); vimar@ptt.rs (V.M.); bb.lena@gmail.com (M.T.-B.); vidana103@gmail.com (A.V.); suvajdzic.nada@gmail.com (N.S.-V.); darko.antic1510976@gmail.com (D.A.); 2Faculty of Medicine, University of Belgrade, 11000 Belgrade, Serbia; 3Faculty of Medicine, Institute for Medical Statistics and Informatics, University of Belgrade, 11000 Belgrade, Serbia; zoran.bukumiric@med.bg.ac.rs

**Keywords:** acute myeloid leukemia, arterial thrombosis, myocardial infarction, stroke, critical limb ischemia

## Abstract

**Simple Summary:**

Patients with hematological malignancies have an increased risk of arterial thrombosis (ATE). Data about incidence and risk factors for ATE development in patients with acute myeloid leukemia (AML) are missing. Out of 626 AML patients, 18 (2.9%) patients developed ATE in the median time of 3 months. Our study showed that the risk of ATE was increased in AML patients with cardiovascular comorbidities, previous thrombosis, adverse cytogenetic risk as well as BMI > 30.

**Abstract:**

Background: Patients with hematological malignancies have an increased risk of arterial thrombotic events (ATEs) after diagnosis, compared to matched controls without cancer. However, data about incidence and risk factors for ATE development in patients with acute myeloid leukemia (AML) are missing. Aim: The objectives of this study were to determine the incidence of ATE in non-promyelocytic-AML patients and to define the potential risk factors for ATE development. Methods: We conducted a retrospective cohort study of adult patients with newly diagnosed AML. The primary outcome was the occurrence of confirmed ATE, defined as myocardial infarction, stroke or critical limb ischemia. Results: Out of 626 eligible AML patients, 18 (2.9%) patients developed ATE in the median time of 3 (range: 0.23–6) months. Half of these patients died due to ATE complications. Five parameters were predictors of ATE: BMI > 30 (*p* = 0.000, odds ratio [OR] 20.488, 95% CI: 6.581–63.780), prior history of TE (*p* = 0.041, OR 4.233, 95% CI: 1.329–13.486), presence of comorbidities (*p* = 0.027, OR 5.318, 95% CI: 1.212–23.342), presence of cardiovascular comorbidities (*p* < 0.0001, OR 8.0168, 95% CI: 2.948–21.800) and cytogenetic risk score (*p* = 0.002, OR 2.113, 95% CI: 1.092–5.007). Conclusions: Our study showed that patients with AML are at increased risk of ATE. The risk was increased in patients with cardiovascular comorbidities, previous thrombosis, adverse cytogenetic risk as well as BMI > 30.

## 1. Introduction

Thromboembolism is an important cause of mortality and morbidity in cancer patients [1,2,3]. While studies regarding venous thrombotic events in cancer patients are numerous, data concerning arterial thrombotic events (ATEs) remain scarce [2,3,4,5]. In the published studies, ATEs have proven to be more common among cancer populations than in non-cancer populations, with an estimated prevalence of between 2–5% [6,7,8,9]. Interestingly, the same cardiovascular risk factors for ATE which are significant in the general population can be applied in patients with cancer [6,8,9,10,11]. However, some cancer-related factors (type and localization of the disease, as well as therapy type) are also associated with a higher risk of ATE [8,9,10,11].

Patients with hematological malignancies have an increased risk of ATE within 6-months (aHR 2.32; 95% CI 2.07–2.60) and 12-months (aHR 1.79; 95% CI 1.63–1.96) after cancer diagnosis, compared to matched controls without cancer [11]. However, high-quality studies on risk factors for ATE in individual subtypes of hematological malignancies are rare. Moreover, the majority of the available data about ATE associated with hematological malignancies concerned multiple myeloma or lymphoma patients, while data regarding patients with acute myeloid leukemia (AML) are anecdotal [12].

Unlike patients with solid tumors, patients with AML are often thrombocytopenic at the time of diagnosis and during the treatment course. Therefore, the prevalence of ATE among those patients was expected to be lower than in those with solid tumors. However, it looks like the prevalence of ATE in AML is as high as in other cancers [13,14]. Data about risk factors for ATE development in AML patients are very rare.

Therefore, the objectives of this study were to determine the incidence of ATE in non-promyelocytic-AML patients and to define the potential risk factors for ATE development.

## 2. Materials and Methods

### 2.1. Patients Characteristics

A retrospective cohort study was conducted in the Clinic of Hematology, University Clinical Center of Serbia. Adult patients (≥18 years of age) with newly diagnosed non-promyelocytic AML between January 2009 and December 2021 were enrolled in the study. The retrieval of information and publication of these results were approved by the Institutional Review Board of the University Clinical Center of Serbia (protocol number III 41004).

AML diagnoses were confirmed by cytological, flow-cytometry, and cytogenetic findings according to the World Health Organization (WHO) and European Leukemia Net criteria [15,16]. Participants were followed from the time of diagnosis to ATE development, death, or six months after the diagnosis. Patients were treated in an intensive (“3 + 7” induction followed by intermediate-dose cytarabine (IDAC) consolidation and allogenic hematopoietic stem-cell transplantation), non-intensive-chemotherapy (azacytidine, low-dose chemotherapy), or supportive manner [16,17] The therapy regimen was chosen according to different parameters, including age, Eastern Cooperative Oncology Group performance status (ECOG PS), Hematopoietic-Cell-Transplantation-specific Comorbidity Index (HCT CI) and disease risk [16,17].

### 2.2. Data Collection

The following baseline parameters were collected: age, sex, body mass index (BMI), smoking status, comorbidities (diabetes mellitus, hypertension (HTA), cardiovascular comorbidities (previous heart attack, coronary heart disease, heart failure, valve disease, stroke, heart rhythm disorders, peripheral arterial disease), previous history of thromboembolisms (TEs), concomitant antiplatelet, anticoagulant or statin therapy on diagnosis, ECOG PS and HCT-CI. Moreover, baseline laboratory parameters such as hemoglobin level, white blood cells, platelets, prothrombin time (PT), activated partial thromboplastin time (APTT), fibrinogen, D-dimer and lactate dehydrogenase (LDH) were included in the analysis. The presence of disseminated intravascular coagulation (DIC) was assessed according to the International Society of Thrombosis and Haemostasis (ISTH) scoring system for DIC [18]. A platelet count > 100 × 10^9^/L accounts for 0 points, a number between 50–100 × 10^9^/L for 1 point and a platelet count < 50 × 10^9^/L for 2 points. Prothrombin time, given in seconds of PT prolongation, results in 0 points when prolongation < 3 s, 1 point when 3–6 s and 2 points when >6 s. Fibrinogen levels > 100 mg/dL result in 0 points, <100 mg/dL in 1 point. No increase in D-dimer levels accounts for 0 points, moderate increase levels for 2 points, and a severe increase accounts for 3 points. A sum score of ≥5 is defined as an overt DIC-2001 score [18]. Diagnosis of leukemia was made according to cytology, flow-cytometry, cytogenetics and molecular-genetics (FLT3, NPM1) studies. Favorable-risk cytogenetics was defined as the presence of inv(16) or t(16;16), t(8;21), as well as normal cytogenetics with nucleophosmin (NPM1) mutations in the absence of an FMS-like tyrosine kinase 3 (FLT3) internal tandem duplication (ITD) mutation, while adverse risk was defined as the presence of complex abnormalities, monosomal karyotype, −5/−5q, −7/−7q, inv(3), t(3;3), t(6;9), t(9;22), or normal cytogenetics with FLT3-ITD mutation. Patients with cytogenetics not classified as favorable or adverse were placed in the intermediate-risk group [15,16]. Finally, the therapy intensity (intensive, non-intensive, supportive therapy) and treatment phase were also taken into account.

The primary outcome was the occurrence of confirmed ATE, defined as myocardial infarction, stroke or critical limb ischemia. Objective evidence for ATE diagnosis included: (1) computed tomography (CT) and/or CT angiography; (2) Doppler-sonography; (3) electrocardiography, echocardiography (e.g., hypokinetic/akinetic and hypotrophic myocardial section without any other existing reason), cardiac biomarkers, and identification of an intracoronary thrombus by angiography. No routine screening for ATE was carried out during the study. Anticoagulant and antiplatelet therapy started before diagnosis of AML were stopped on admission day and managed according to platelet count, e.g., therapy was restarted if the platelet count was >50 × 10^9^/L. Patients with newly diagnosed ATE during follow-up were treated with antiplatelet therapy if the platelet count was >30 × 10^9^/L.

### 2.3. Statistical Analysis

Methods of descriptive and analytical statistics were used. In statistical analysis, we used all eligible cases without imputation. The normality of data distribution was assessed by histogram and Kolmogorov–Smirnov test. Continuous variables following normal distribution are presented as mean ± standard deviation (SD), while those not following normal distribution are presented as median (range). Categorical variables are shown as absolute (relative) frequencies and were analyzed by the Chi-square test or Fisher’s exact test. The Mann–Whitney–Wilcoxon or Kruskal–Wallis test by rank was applied for continuous variables where appropriate. Univariate logistic regression analyses were used to calculate the risk factors for thrombosis development. The significance level was set at 0.05. Statistical data analysis was performed using IBM SPSS Statistics 22 (IBM Corporation, Armonk, NY, USA).

## 3. Results

A total of 626 consecutive patients with newly diagnosed AML were treated at our center between January 2009 and December 2021. The mean age of the participants was 55.1 ± 13.4 years, and 348 (55.6%) were males. A prior history of TE was recorded in 42/619 (6.8%) patients, and the majority of them were ATE (acute myocardial infarction 23, stroke 11). Cardiovascular comorbidities were registered in 131/614 (21.3%) patients. Diabetes mellitus and HTA were registered in 102/586 (17.4%) and 156/586 (26.6%), respectively. At the moment of hospitalization, 47 (7.5%) patients were on anticoagulant, 31/626 (4.9%) on antiplatelet and 36/610 (5.9%) on statin therapy. During 6 months of follow-up, bleeding events were registered in 260/626 (41.5%) patients and grades 3 and 4 in 83/626 (13.3%) patients. Patients’ and disease characteristics are shown in Table 1.

### Arterial Thrombosis Events

During the follow-up, 18/626 (2.9%) patients developed ATE: ischemic stroke 12/18 (66.7%), myocardial infarction 5/18 (27.8%), and acute lower extremity arterial thrombosis 1/18 (5.5%). ATE was diagnosed most commonly during the induction (9 (50%) patients), reinduction (3 (16.7%) patients) and consolidation (4 (22.2%) patients). Moreover, cases of ATE were noted at diagnosis (1 (5.6%) patient) and after transplantation (1 (5.6%) patient). The median time to thrombosis was 3 (range: 0.23–6) months. Half of the patients died to ATE complications. The characteristics of patients with arterial thrombosis are shown in Table 2.

Among the 38 tested parameters, five were predictors of arterial thrombosis: BMI > 30 (*p* = 0.000, odds ratio [OR] 20.488, 95% CI: 6.581–63.780), prior history of TE (*p* = 0.041, OR 4.233, 95% CI: 1.329–13.486), presence of comorbidities (*p* = 0.027, OR 5.318, 95% CI: 1.212–23.342), presence of cardiovascular comorbidities (*p* < 0.0001, OR 8.0168, 95% CI: 2.948–21.800) and cytogenetic risk score (*p* = 0.002, OR 2.113, 95% CI: 1.092–5.007). Comparisons of patient and disease parameters are shown in Table 1.

## 4. Discussion

We analyzed the incidence of ATE in AML patients and identified patients and disease-related risk factors for ATE development. During a follow-up period of 6 months, 2.9% of patients developed ATE, defined as myocardial infarction, stroke or peripheral artery disease. The risk of ATE was increased in patients with a higher BMI, higher number of cardiovascular comorbidities, prior history of TE and high cytogenetic risk score. The mortality rate due to ATE was 50%. The negative impact of ATE on therapy, prognosis and survival indicated an unmet need for strategies to prevent, treat and manage ATE. However, prophylaxis and management are challenging due to universally presented thrombocytopenia and systemic coagulopathy. In circumstances of very high bleeding, risk administration of antithrombotic therapy can be fatal. Consequently, determining ATE development risk factors will allow clinicians to individualize patient surveillance and prophylaxis. However, according to our knowledge, this is the first study aiming to explore the frequency and risk factors related to ATE in patients with AML.

The frequency of ATE in our study (2.9%) is in line with previously published data for AML (2.3%) and cancer patients (2–7%) [6,7,8,9,10,11,12,13,14]. Cancer-related studies have noted an increased risk of ATE in the period preceding cancer diagnosis as well as during the first months following diagnosis, and a quick decline thereafter [6,7,8,9,10,11,12]. In our group of patients, ATE most frequently developed in the first month after diagnosis, typically in patients with active disease (5.6% on diagnosis, 50% during induction and 6.7% during reinduction). This data suggests that AML is independently associated with a higher risk of ATE, irrespective of conventional cardiovascular risk factors. The mechanism of increased thrombotic events in cancers remains debated. It is known that acute leukemia has inherent procoagulant properties by releasing tissue factor, cancer procoagulant and tumor necrosis factor as well as interleukin 1 [12,19,20,21]. Growing evidence has shown that cancer treatment can worsen patients’ thrombophilic state via direct endothelial damage, the destruction of leukemic cells releasing thrombogenic substances, and the reduction of the synthesis of natural anticoagulants due to liver damage [20,21].

Our study investigated the predictive value of 38 patient-, disease- and therapy-related parameters for ATE development. The risk factors identified as predictive in our study are consistent with previous published data for cancer patients [6,7,8,9,10,11,12]. The same patient-related risk factors for ATE identified in a no-cancer population can be applied to patients with cancer. Conventional cardiovascular risk factors, such as male sex, age, diabetes mellitus, hypertension, dyslipidemia, smoking, as well as a previous history of thrombosis and use of antiplatelet or anticoagulant therapy (possibly as a surrogate of pre-existing cardiovascular disease/risk), were associated with ATE in a general population and cancer patients [6,7,8,9,10,11,12]. The history of ATE had the highest association (aHR 2.96; 95% CI 2.77–3.17) [11,13]. Among our patient-related risk factors, BMI > 30, prior history of TE, presence of comorbidities and cardiovascular comorbidities as well as adverse genetic risk group were predictive for ATE in univariate analysis. Surprisingly, age did not emerge as a predictive factor, probably due to the younger age of the patients included in our group.

With the exception of patient-related cardiovascular risk factors, previous studies identified a few cancer-related risk factors for ATE, including: cancer type, regional and distant disease (vs. localized), chemotherapy, radiotherapy or surgery (vs. no therapy) [6,7,8,9,10,11,12,13]. An elevated absolute neutrophil count and higher soluble P-selectin levels were identified as potential laboratory risk factors [22]. Our univariate analyses of disease-related parameters identified an adverse genetic risk score as being predictive of ATE. Martella et al.’s and Hellou et al.’s studies, which explored the risk factor for VTE in AML patients, described patients with intermediate-risk cytogenetics as having an increased risk for venous thrombosis [14,23]. In contrast, Lee YG’s study showed that advanced cytogenetic risk was an independent predictor for VTE development [16]. The author of this study concluded that patients in the high-risk group are more likely to have received salvage re-induction chemotherapy. The frequent use of intense chemotherapy in the high-risk cytogenetic group may be related to the predictive role of cytogenetic risk in VTE development [16]. However, the majority of thromboses in our group were diagnosed during the induction cycle. Calvillo-Argüelles et al.’s study, which investigated the link between CHIP-related mutations in AML patients and the rate of cardiovascular events, showed that the presence of any CHIP-related mutation (DNMT3A, TET2, ASXL1, TP53, JAK2, SRSF2, and SF3B1) was associated with a higher rate of thrombosis. Furthermore, TP53 and ASXL1 mutations were associated with the occurrence of VTEs [24]. In Kattih at al.’s paper, the presence of IDH1 mutations was associated with a higher rate of coronary artery disease [25]. This data can suggest that some molecular subgroups of patients are predisposed to cardiovascular disease and should be closely monitored from a cardiovascular standpoint during treatment [26]. However, at this moment, the cause of the relationship between ATE and the cytogenetic relation is unknown to us.

Our findings raise the question of whether patients with newly diagnosed AML, particularly those with advanced cytogenetic and cardiovascular comorbidities and high BMI, should be considered for primary prevention of cardiovascular disease. A phase I trial evaluating the efficacy of aspirin with or without statins in thrombosis prevention in solid tumor patients at a high or intermediate risk for ATE might provide us with answers to the question on the rational use of antiplatelet drugs for primary ATE prevention (clinicaltrials.gov identifier:02285738) [27]. However, bleeding is a very frequent complication during AML treatment, with 13.3% patients developing grade 3 or 4 bleeding events during follow-up in our group. Consequently, an inherent risk of bleeding may offset the benefits of prophylactic aspirin usage. Therefore, we need tools for the assessment of the bleeding risk in AML, together with an ATE development risk tool. In the meantime, AML-patient-treating doctors should continue with standard management strategies, as in the case of the non-cancer population (manage general cardiovascular risk factors, such as obesity, hypertension and hyperlipidemia, and be vigilant of symptoms or signs of ATE) [12,28].

This study had limitations. First, this study used a retrospective analysis, and the underlying bias could not be avoided. However, we included all consecutive patients diagnosed at our center, thus reducing the risk of selection bias. Due to the retrospective nature of the study data, parameters are missing for some patients. Secondly, our hospital is an academic center, mainly treating patients qualified for intensive therapy, and thus our population probably does not reflect the full spectrum of patients with AML. Furthermore, the presented data are driven from the relatively small Serbian population, which lacks diversity in view of race and ethnicity. Finally, since the number of ATE events was small, multivariate analyses with a predictive model development could not be driven from these data.

## 5. Conclusions

Our study showed that patients with AML are at an increased risk of arterial thromboembolic events, including myocardial infarction and stroke. The risk of ATE was increased in patients with BMI > 30, cardiovascular comorbidities, previous thrombosis and adverse cytogenetic risk. In conclusion, there is a growing need for dedicated primary thromboprophylaxis trials in patients with acute leukemia in a risk-adapted manner.

## Figures and Tables

**Table 1 cancers-15-03060-t001:** Patients and disease-related characteristics in acute myeloid leukemia patients with and without thrombosis.

Parameter		All N = 626	Patients with Thrombosis N = 18	Patients without Thrombosis N = 608	OR	95% CI	*p*
Age (mean years)		55.1	59.2	55.0	1.028	0.986–1.071	0.187
Male sex		348/626 (55.6%)	7/18 (38.9%)	341/608 (56.1%)	0.498	0.191–1.303	0.148
Smokers		277/592 (46.8%)	8/17 (47.1%)	269/575 (46.8%)	1.011	0.385–2.658	0.982
BMI ^1^ > 30		96/580 (16.6%)	14/18 (77.8%)	82/562 (14.6%)	20.488	6.581–63.780	<0.001
Prior history of thrombotic events		42/619 (6.8%)	4/18 (22.2%)	38/601 (6.3%)	4.233	1.329–13.486	0.028
ECOG PS ^2^	0	102/611 (16.7%)	3/18 (16.7%)	99/593 (16.7%)	0.987	0.610–1.597	0.958
	1	256/611 (41.9%)	6/18 (33.3%)	250/593 (42.2%)			
	2	182/611 (29.8%)	8/18 (44.4%)	174/593 (29.3%)			
	3	48/611 (7.9%)	1/18 (5.6%)	47/593 (7.9%)			
	4	23/611 (3.9%)	0/18	23/593 (3.8%)			
Comorbidities	Any	374/614 (60.9%)	16/18 (88.9)	358/596 (60.1%)	5.318	1.212–23.342	0.027
	Diabetes	102/586 (17.4%)	3/16 (18.8%)	99/570 (17.4%)	1.098	0.307–3.925	0.886
	Hypertension	156/586 (25.0%)	4/16 (25.0%)	152/570 (26.7%)	0.917	0.291–2.886	0.882
	Cardiovascular	131/614 (21.3%)	12/18 (66.7%)	119/596 (20.0%)	8.0168	2.948–21.800	<0.0001
Concomitant therapy on diagnosis	Statins	36/615 (5.9%)	2/18 (11.1%)	34/597 (5.7%)	0.483	0.107–2.187	0.345
	Antiplatelet	33/614 (5.4%)	2/18 (11.1%)	31/596 (5.2%)	0.439	0.097–1.994	0.286
	Anticoagulant	50/615 (8.1%)	3/18 (16.7%)	47/597 (7.9%)	0.427	0.119–1.529	0.191
HCT CI ^3^	Low (0–2) High (>2)	466/611 (76.3%) 145/611 (23.7%)	12/17 (70.6%) 5/17 (29.4%)	454/594 (76.4%) 140/594 (23.6%)	0.740	0.256–2.137	0.568
^4^ CNS involvement		54/264 (20.5%)	0/9 (0.0%)	54/255 (21.2%)	–	–	0.174
^5^ WBC (median, normal: 3.6–10 × 10^9^/L)		9.8	9.5	9.8	0.994	0.983–1.006	0.955
Platelet count (median, normal: 150–400 × 10^9^/L)		49	32	49	1.001	0.995–1.007	0.211
Hemoglobin (mean, normal: 120–160 g/L)		95.8	98.2	95.8	1.008	0.982–1.034	0.708
^6^ LDH (mean, normal)		458	468	458	1.000	1.000–1.001	0.476
Fibrinogen (median, normal: 2.2–5.5 g/L)		5.4	5.2	5.4	1.022	0.914–1.143	0.560
^7^ iNR (mean, normal: 0.8-1.3%)		1.22	1.21	1.22	0.739	0.052–10.557	0.924
^8^ APTT (mean, normal: 25.1–36.5 s)		29.2	27.4	29.2	0.914	0.811–1.030	0.511
D dimer (median, normal: 0–0.5 μg/L)		2.5	3.0	2.5	0.990	0.947–1.035	0.824
^9^ ISTH DIC criteria	0–4 >4	189/328 (57.6%) 139/328 (42.4%)	3/9 (33.3%) 6/9 (66.7%)	186/319 (58.3%) 133/319 (41.7%)	0.368	0.090–1.496	0.176
Blast peripheral blood (median, %)		16	35	16	1.009	0.994–1.023	0.414
Cytogenetic risk group (N)	Favorable	63/555 (11.4%)	3/16 (18.8%)	60/539 (11.1%)	2.113	1.092–5.007	0.002
	Intermediate	330/555 (59.5%)	3/16 (18.8%)	327/539 (60.7%)			
	Advance	162/555 (29.2%)	10/16 (62.5%)	152/539 (28.2%)			
^10^ CVL inserted (N)		519/626 (82.9%)	15/18 (83.3%)	504/608 (82.9%)	1.032	0.293–3.628	0.961
Therapy type	Intensive	453/626 (72.4%)	11/18 (61.1%)	442/608 (72.7%)	0.590	0.225–1.548	0.284
	Non-intensive/supportive	173/626 (27.6%)	7/18 (38.9%)	166/608 (27.3%)			

^1^ BMI—body mass index; ^2^ ECOG-PS—Eastern Cooperative Oncology Group performance status; ^3^ HCT CI—Hematopoietic cell transplantation-specific comorbidity index; ^4^ CNS—central nervous system; ^5^ WBC—white blood cell; ^6^ LDH—Lactate dehydrogenase; ^7^ INR—international normalized ratio; ^8^ aPTT—activated partial thromboplastin time; ^9^ ISTH DIC criteria—International Society of Thrombosis and Haemostasis criteria for disseminated intravascular coagulation; ^10^ CVL—central venous line.

**Table 2 cancers-15-03060-t002:** Characteristics of arterial thrombotic events.

Patient	Age	Sex ^1^	Comorbidities	BMI ^2^	Cytogenetic Risk Group	Thrombosis Localization	Time to Thrombosis (Months)	Death Due to Thrombosis
1	65	m	St.post. MI ^3^, St. post. oclusio a. centralis retinae, basocellular skin cancer, MPN ^4^	20.65	Adverse	CVI	4	Yes
2	48	f	HTA ^5^	28.04	Adverse	CVI	3.5	Yes
3	67	m	HTA, HF	29.10	Intermediate	MI	3	No
4	60	m	HTA, DM ^6^ type 2, testicular cancer, basocellular scin cancer	34.00	Adverse	Critical limb ischemia	1	No
5	69	f	Obesity, HTA, HF	40.20	Adverse	CVI	5	No
6	69	f	AF	21.58	Intermediate	CVI	2	Yes
7	62	f	HTA, HF	24.50	Adverse	CVI	3	Yes
8	62	m	HTA, AF	26.70	Intermediate	MI	0.5	yes
9	72	f	HTA, st. post CVI ^7^	29.30	Adverse	CVI	5	yes
10	52	f	HTA	30.20	Adverse	CVI	3	no
11	36	f	/	20.70	Adverse	CVI	6	no
12	53	m	/	20.00	Adverse	CVI	4	no
13	63	m	/	20.50	Adverse	CVI	2	yes
14	36	m	/	23.70	Favorable	MI	3	no
15	63	f	Depression, chronic, obstructive lung disease, rheumatoid arthritis, St. post MI	25.00	Favorable	CVI	1	no
16	56	f	HTA	25.30	Not assessed	MI	0.3	yes
17	72	F	HTA, DM type 2, diabetes insipidus, Status post CVI	32.00	Adverse	CVI	5	yes
18	60	f	HTA, AF, Hashimoto thyroiditis	26.80	Not assessed	MI	0.2	yes

^1^ Sex: m—male, f—female; ^2^ BMI—body mass index; ^3^ MI—myocardial infarction; ^4^ MPN—myeloproliferative neoplasm; ^5^ HTA—hypertension; ^6^ DM—diabetes mellitus; ^7^ CVI—cerebrovascular insult, stroke, HF heart failure, AF atrial fibrillation.

## Data Availability

All data regarding this research are available upon reasonable request to the corresponding author.

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
