# Peer review of "Arterial Thrombosis in Patients with Acute Myeloid Leukemia: Incidence and Risk Factors"

_cancers, 2023, doi:10.3390/cancers15113060_

Round 1

Reviewer 1 Report

The article by Mitrovic et al. investigates the incidence and risk factors of arterial thrombotic events (ATE) in patients with acute myeloid leukemia (AML). The authors find that ATE occurred in 2.9% of AML patients with a median time to event of 3 months. They identify five predictors of ATE: BMI>30, prior hystory of TE, comorbidities, in particular cardiovascular comorbidities, and adverse ELN risk score. The article is well-organized, clear and concise. However, i suggest several minor changes:

Line 95: specify how ISTH-DIC score 2018 was calculated

Line 131 and table 1:I suggest to use a cut-off for the number of comorbidities rather than a median value, or use the Charlson comorbidity index.

Table 1: For ISTH DIC score, i suggest to use the cut-off of 4, as indicated by the ISTH to indicate "overt" DIC, to compare the two groups instead of the median value. Same for the HCT-CI. I suggest also to exclude the cytofluorimetric parametres from the table and moved to a supplementary materials including also patient's cytogenetic features.

It would be of interest to further discuss the correlation between treatment intensity and the development of thrombosis

Line 16: i would suggest "not conclusive" or "anecdotal" instead of "missing". Same at line 24

Table 1, Line 172 and 212: i suggest to use "favorable" and "adverse" ELN risk instead of "good" and "high"

Author Response

Question: Line 95: specify how ISTH-DIC score 2018 was calculated.
Answer: A platelet count >100x109/L accounts for 0 points, a score between 50-100x109/L for 1 point and a platelet count <50x109/L for 2 points. Prothrombin time, given in seconds of PT prolongation, results in 0 points when prolongation <3s, 1 point when 3-6 s and 2 points when >6 s. Fibrinogen levels >100 mg/dL result in 0 points, <100 mg/dL in 1 point. No increase D-dimer levels account for 0 points, moderate increase levels for 2 points and sever increase account for 3 points. A sum score of ≥ 5 is defined as an overt DIC-2001 score.

Question: Line 131 and table 1: I suggest to use a cut-off for the number of comorbidities rather than a median value, or use the Charlson comorbidity index
Answer: Thank you for your suggestion. We replaced median value with presence of any comorbidities, cardiovascular comorbidities as well as cut-off values for DIC score and HCT- CI comorbidity index. We are using Hematopoietic Cell Transplantation-specific Comorbidity Index in every day practice rather than Charlson comorbidity index

Question: For ISTH DIC score, I suggest to use the cut-off of 4, as indicated by the ISTH to indicate "overt" DIC, to compare the two groups instead of the median value. Same for the HCT-CI.
Answer: Thank you for suggestion. We replaced median values with cut-off values.

Question:  I suggest also to exclude the cytofluorimetric parameters from the table and move to a supplementary materials including also patient's cytogenetic features.
Answer: Thank you for suggestion. Cytofluorimetric and cytogenetic parameters were removed to a supplementary material.

Question: Line 16: I would suggest "not conclusive" or "anecdotal" instead of "missing". Same at line 24. Table 1, Line 172 and 212: I suggest to use "favorable" and "adverse" ELN risk instead of "good" and "high"
Answer: Thank you for suggestion. We corrected the text.

Reviewer 2 Report

Mitrovic and colleagues report a rather large retrospective study exploring the incidence and risk factors of ATE in AML patients. While the report could be of interest, some points of concern should be addressed:

1      How were missing values handled? The % reported in the table describing patient’s characteristics are likely derived excluding missing values, but this should be clarified. I suggest explaining this aspect in the statistical methods and to explicit numbers in the table. This limit should also be discussed.

2    You refer to ELN risk classification, and you cite the 2022. However, your risk stratification is not consistent, no NGS molecular data are mentioned (thus preventing a 2022 classification), and there is confusion between cytogenetic and molecular risk. Please clarify this point and chose a risk stratification you can actually apply.

3    While the cited literature is globally accurate, some relevant papers, especially focused on the possible molecular predictors of cardiovascular event are missing (eg Kattih B,. Leukemia. 2021;35(5):1301–16.;   Calvillo-Argüelles O, Schoffel A, Capo-Chichi J-M,. JACC CardioOncol. 2022;4(1):38–49). Indeed, these aspects should be further discussed. Besides, recent reviews could also be helpful (eg, Olivi, Current Treatment Options in Oncology 2023)

4    In the introduction, you state: “there are no data about risk factors for ATE development in AML”. Indeed, this is not true, as you clearly explained in the discussion. Please rectify.

5     You state de novo AML were included. Thus, do you mean you excluded secondary/therapy rel AML? Why? Or did you mean newly diagnosed

6  Surprisingly, age did not emerge as a predictive factor. I suggest to briefly comment this aspect

7     You report 20% CNS involvement in newly diagnosed AML. This is surprising, as these cases are usually extremely rare. How do you explain this data?

 8 The layout of the tables should be improved, as some data are redundant (eg all the CD markers). Besides, the variable should be better described (eg, avoid “average age”)

English is globally clear, but some mistakes (eg, missing article in line 57), repetitions (eg line 45) are present. Thus, it should be improved

Author Response

Question: How were missing values handled? The % reported in the table describing patient’s characteristics are likely derived excluding missing values, but this should be clarified. I suggest explaining this aspect in the statistical methods and to explicit numbers in the table. This limit should also be discussed.
Answer:  Thank you for question. In the method section we added: In statistical analysis we used all eligible cases without imputation. In the table we added the numbers of patients with data. Also in the study limitations we added: Due to retrospective nature of the study data for some parameters are missing.

Question: You refer to ELN risk classification, and you cite the 2022. However, your risk stratification is not consistent, no NGS molecular data are mentioned (thus preventing a 2022 classification), and there is confusion between cytogenetic and molecular risk. Please clarify this point and chose a risk stratification you can actually apply.
Answer:  Thank you for this remarkable observation. Since 2009, the classifications have changed. The possibility of our center in terms of diagnostics was also changing. Therefore, we reformulated the classification into: Risk grouping of patients was based on initial cytogenetics and molecular abnormalities. Patients with inv(16) or t(16;16), t(8;21), t(15;17), or normal cytogenetics with nucleophosmin (NPM1) mutations in the absence of an FMS-like tyrosine kinase 3 (FLT3) internal tandem duplication (ITD) mutation were classified as belonging to a low cytogenetic risk group. Patients with complex abnormalities, monosomal karyotype, –5/-5q, –7/-7q, inv(3), t(3;3), t(6;9), t(9;22), or normal cytogenetics with FLT3-ITD mutations were classified as the high-risk group. The remaining patients were classified as the intermediate-risk group.

Question: While the cited literature is globally accurate, some relevant papers, especially focused on the possible molecular predictors of cardiovascular event are missing (eg Kattih B,. Leukemia. 2021;35(5):1301–16.;  Calvillo-Argüelles O, Schoffel A, Capo-Chichi J-M,. JACC CardioOncol. 2022;4(1):38–49). Indeed, these aspects should be further discussed. Besides, recent reviews could also be helpful (eg, Olivi, Current Treatment Options in Oncology 2023)
Answer: Thank you for suggestion. We incorporate suggested papers in reference list. In discussion we added: Calvillo-Argüelles et al study, investigated the link between CHIP-related mutations in AML patients and the rate of cardiovascular events, showed that presence of any CHIP-related mutation (DNMT3A, TET2, ASXL1, TP53, JAK2, SRSF2, and SF3B1) was associated with a higher rate of thrombosis. Besides, TP53 and ASXL1 mutations were associated with the occurrence of VTEs. In the Kattih at al paper the presence of IDH1 mutations was associated with a higher rate of coronary artery disease. This data can suggest that some molecular subgroups of patients are predisposed to cardiovascular disease and should be closely monitored from a cardiovascular standpoint during treatment.

Question: In the introduction, you state: “there are no data about risk factors for ATE development in AML”. Indeed, this is not true, as you clearly explained in the discussion. Please rectify.
Answer: Thank you for suggestion. We corrected to: Data about risk factors for ATE development in AML patients are very rare.

Question: You state de novo AML were included. Thus, do you mean you excluded secondary/therapy rel AML? Why? Or did you mean newly diagnosed
Answer: We included all newly diagnosed patient, we included secondary/therapy related AML. In the method section we corrected it in newly diagnosed AML patients.

Question: Surprisingly, age did not emerge as a predictive factor. I suggest to briefly comment this aspect
Answer: We added in the discussion: Surprisingly, age did not emerge as a predictive factor, probably due to younger age of the patients included in our group.

Question: You report 20% CNS involvement in newly diagnosed AML. This is surprising, as these cases are usually extremely rare. How do you explain this data?
Answer: Thank you for this interesting question. In our center, analysis of cerebrospinal fluid is not limited only to patients with CNS symptoms. The analysis is also performed on patients with hyperleukocytosis, FAB types AML M4 and 5 as well as those who are CD56 positivity.  In this selected group we had high percentage of patients with CNS involvement. However, majority of patients was not tested. We corrected this in table now.  

Question: The layout of the tables should be improved, as some data are redundant (eg all the CD markers). Besides, the variable should be better described (eg, avoid “average age”)
Answer: Thank you for suggestion. We corrected table and removed some data in supplement.

Question: English is globally clear, but some mistakes (eg, missing article in line 57), repetitions (eg line 45) are present. Thus, it should be improved
Answer: Thank you for suggestion. We corrected the text.

Reviewer 3 Report

Comments to the Author

The manuscript entitled “Arterial thrombosis in patients with acute myeloid leukemia: incidence and risk factors” assessed incidence and risk factors for ATE in adults with AML. The primary finding is that approximately 3% of patients with AML had an ATE within 6 months of diagnosis, those with cardiovascular comorbidities, previous thrombosis, adverse European LeukemiaNet stratification and high BMI >30 having higher risk. The manuscript is well written and concise.

The authors have compared 42 parameters. I am not quite sure which ones, in Table 1 there are 38 comparisons/p-values. Multiple testing increases the risk of type I error (such as for FLT3 ITD positivity, where positive FLT3 seems to increase chances of not getting an ATE).

Specific very minor comments for the authors are outlined below:

Anticoagulant or antiplatelet therapy at diagnosis was not a significant risk factor, even if they were routine stopped at diagnosis. This is somewhat unexpected, for example one would have expected those with atrial fibrillation without anticoagulation to have higher risk for stroke, but this subanalysis was not possible doe to small cohort or there was no evidence for that in your study?

Also, the authors mention that patients with ATE treated with antiplatelet therapy when the platelet count was >30. Please clarify, does this refer to those with previous ATE (as the study time/follow-up ended at the event of ATE)

Cardiovascular comorbidities were rare. Which cardiovascular comorbidities were included (peripheral arterial disease, coronary heart disease, history of stroke?)

Some abbreviations are missing in the main text (such as HTA).

ELN stratification is classified in Table 1 Good/Intermediate/High and in Table 2 Favourable/Intermediate/Adverse

Author Response

Question: The authors have compared 42 parameters. I am not quite sure which ones, in Table 1 there are 38 comparisons/p-values.
Answer: Thank You for suggestion. We counted each comorbidity and concomitant therapy separately. We corrected it to 38.

Question: Anticoagulant or antiplatelet therapy at diagnosis was not a significant risk factor, even if they were routine stopped at diagnosis. This is somewhat unexpected, for example one would have expected those with atrial fibrillation without anticoagulation to have higher risk for stroke, but this subanalysis was not possible doe to small cohort or there was no evidence for that in your study?
Answer: Thank You for this question. It was surprising for us that the use of antiplatelet and anticoagulant therapy was not predictive. Although we do not have a firm explanation, it is most likely due to a small number of events. The number of patient with AF was to small for subanalysis.

Question: Also, the authors mention that patients with ATE treated with antiplatelet therapy when the platelet count was >30. Please clarify, does this refer to those with previous ATE (as the study time/follow-up ended at the event of ATE)
Answer: We added to the text: Patients with newly diagnosed ATE during follow-up were treated with antiplatelet therapy if the platelet count was >30x109/L.

Question: Cardiovascular comorbidities were rare. Which cardiovascular comorbidities were included (peripheral arterial disease, coronary heart disease, history of stroke?)
Answer: We include previous heart attack, coronary heart disease, heart failure, valve disease, stroke, heart rhythm disorders, peripheral arterial disease). Low number of comorbidities can be explained by younger age of patients across our cohort.  

Question: Some abbreviations are missing in the main text (such as HTA).
Answer: Thank you. We corrected it.

Question: ELN stratification is classified in Table 1 Good/Intermediate/High and in Table 2 Favourable/Intermediate/Adverse
Answer: Than You for the suggestion, we made a correction.

Round 2

Reviewer 2 Report

the authors adequately addressed my concerns